Molecular serotyping of Haemophilus parasuis isolated from diseased pigs and the relationship between serovars and pathological patterns in Taiwan

Lin Wei-Hao 1 2
Shih Hsing-Chun 2
Lin Chuen-Fu 3
Yang Cheng-Yao 4
Chang Yung-Fu 5
http://orcid.org/0000-0003-1911-7535 Lin Chao-Nan 1 2 cnlin6@mail.npust.edu.tw
Chiou Ming-Tang 1 2 6 mtchiou@mail.npust.edu.tw
1 Department of Veterinary Medicine, College of Veterinary Medicine, National Pingtung University of Science and Technology , Pingtung , Taiwan
2 Animal Disease Diagnostic Center, College of Veterinary Medicine, National Pingtung University of Science and Technology , Pingtung , Taiwan
3 Department of Veterinary Medicine, College of Veterinary Medicine, National Chiayi University , Chiayi , Taiwan
4 Graduate Institute of Veterinary Pathobiology, College of Veterinary Medicine, National Chung Hsing University , Taichung , Taiwan
5 Department of Population Medicine and Diagnostic Sciences, College of Veterinary Medicine, Cornell University , NY , USA
6 Research Center for Animal Biologics, National Pingtung University of Science and Technology , Pingtung , Taiwan
Blackburn Jason
Electronic publication date: 2018 Nov 29
Publication date: 2018
Volume: 6
Electronic Location ID: e6017
Received 2018 Aug 28; Accepted 2018 Oct 26
Copyright: © 2018 Lin et al.
Copyright year: 2018
Copyright holder: Lin et al.
License: This is an open access article distributed under the terms of the Creative Commons Attribution License, which permits unrestricted use, distribution, reproduction and adaptation in any medium and for any purpose provided that it is properly attributed. For attribution, the original author(s), title, publication source (PeerJ) and either DOI or URL of the article must be cited.
License URL: https://creativecommons.org/licenses/by/4.0/

Keywords: Glässer’s disease, Haemophilus parasuis, Polyserositis, Serotyping

Funding: Animal Disease Diagnostic Center College of Veterinary Medicine National Pingtung University of Science and Technology, Taiwan This study was supported by the Animal Disease Diagnostic Center, College of Veterinary Medicine, National Pingtung University of Science and Technology, Taiwan. The funders had no role in study design, data collection and analysis, decision to publish, or preparation of the manuscript.

==============================
Background

Haemophilus parasuis is the etiological agent of Glässer’s disease, and causes severe economic losses in the swine industry. Serovar classification is intended as an indicator of virulence and pathotype and is also crucial for vaccination programs and vaccine development. According to a polysaccharide biosynthesis locus analysis, H. parasuis isolates could be classified by a molecular serotyping assay except serovars 5 and 12 detected by the same primer pair. The aim of this study was to identify H. parasuis isolates from diseased pigs in Taiwan by using a molecular serotyping assay and to analyze the relationship between serovars and pathological patterns.

Methods

From August 2013 to February 2017, a total of 133 isolates from 277 lesions on 155 diseased animals from 124 infected herds serotyped by multiplex PCR and analyzed with pathological data.

Results

The dominant serovars of H. parasuis in Taiwan were serovars 5/12 (37.6%), 4 (27.8%) and 13 (15%) followed by molecular serotyping non-typable (MSNT) isolates (13.5%). Nevertheless, the serovar-specific amplicons were not precisely the same sizes as previously indicated in the original publication, and MSNT isolates appeared with unexpected amplicons or lacked serovar-specific amplicons. Most H. parasuis isolates were isolated from nursery pigs infected with porcine reproductive and respiratory syndrome virus. The percentage of lung lesions (30.4%) showing H. parasuis infection was significantly higher than that of serosal lesions.

Discussion

Collectively, the distribution of serovars in Taiwan is similar to that found in other countries, but MSNT isolates remain due to genetic variations. Furthermore, pulmonary lesions may be optimum sites for H. parasuis isolation, the diagnosis of Glässer’s disease, and may also serve as points of origin for systemic H. parasuis infections in hosts.

Introduction

Haemophilus parasuis, a part of normal upper respiratory microbiota, is the etiological agent of Glässer’s disease which induces sudden death, polyserositis, polyarthritis, meningitis and poor production performance, resulting in severe economic losses in the swine industry (Amano et al., 1994; Moller & Kilian, 1990; Vahle, Haynes & Andrews, 1997; Zhang et al., 2014). Vaccination is an effective strategy for preventing increased mortality and economic losses caused by virulent H. parasuis (Miniats, Smart & Ewert, 1991a; Smart & Miniats, 1989). However, only partial protection is observed with heterologous H. parasuis challenges due to poor cross-protection (Miniats, Smart & Rosendal, 1991b; Nielsen, 1993; Takahashi et al., 2001). Thus, serotyping of H. parasuis is very important, not only for epidemiological research but also for choosing efficacious inactivated whole-cell bacterial vaccines.

A total of 15 serovars, conventional serotyping cross-reactive (CSCR) and conventional serotyping non-typable (CSNT) isolates of H. parasuis have been described and demonstrated by gel immunodiffusion assay (GID) (Kielstein & Rapp-Gabrielson, 1992). Due to the persistence of cross-reactivity or non-reaction to antisera, there are still approximately 15–40% CSCR and CSNT isolates reported in a variety of countries by GID (Table S1) (Blackall, Rapp-Gabrielson & Hampson, 1996; Cai et al., 2005; Castilla et al., 2012; Del Rio, Gutierrez & Rodriguez Ferri, 2003; Kielstein & Rapp-Gabrielson, 1992; Luppi et al., 2013; Ma et al., 2016; Oliveira, Blackall & Pijoan, 2003; Rapp-Gabrielson & Gabrielson, 1992; Rubies et al., 1999; Tadjine et al., 2004). Despite using an indirect hemagglutination assay designed to reduce the proportion of CSCR isolates, 7.5–18% of isolates are still untypable (Table S1) (Angen, Svensmark & Mittal, 2004; Cai et al., 2005; Del Rio, Gutierrez & Rodriguez Ferri, 2003; Dijkman et al., 2012; Howell et al., 2015). This phenomenon makes it more difficult to conduct an effective vaccination program against H. parasuis.

Conventional serotyping is used extensively (Kielstein, Rosner & Mueller, 1991; Morozumi & Nicolet, 1986; Rapp-Gabrielson & Gabrielson, 1992). The Kielstein-Rapp-Gabrielson scheme recognizes 15 serovars of H. parasuis on the basis of a GID test with specific rabbit antisera and the authors noted a correlation between serovar and virulence (Kielstein & Rapp-Gabrielson, 1992). According to serotyping results, serovar 4 tends to be found in pulmonary infections; CSNT and CSCR isolates are mainly found in systemic infections (Angen, Svensmark & Mittal, 2004). Unfortunately, others report little correlation between serovar and virulence as isolates in the same serovar often exhibit different virulence levels (Aragon et al., 2010; Olvera, Segales & Aragon, 2007).

Previous studies established the serovar and pathotype of H. parasuis are based on differences at the genome level (Brockmeier et al., 2014; Howell et al., 2013, 2017). A multiplex PCR (mPCR) based on a polysaccharide biosynthesis locus analysis was employed to molecularly serotype H. parasuis serovars (Howell et al., 2015). As a result, 14 of 15 serovars of H. parasuis (serovars 5 and 12 could not be differentiated) were identified using this assay (Howell et al., 2015). Using the molecular typing assay many of the CSNT and CSCR isolates were successfully typed in a recognized serovar.

Although Glässer’s disease is common in Taiwan, serotyping of pathogenic H. parasuis isolates from Taiwanese pigs is not clear. The principal aim of this study was to molecularly serotype H. parasuis isolated from Taiwanese diseased pigs, and correlate serovars with pathological patterns.

Materials and Methods

Bacterial isolate collection and identification

Haemophilus parasuis field isolates were collected from diseased pig herds between August 2013 and February 2017 in Taiwan (Table S2). Lesions suspected of being caused by H. parasuis in diseased pigs were located in the meninges, pleura, pericardia, peritonea, synovial cavities of joints and lungs. Lesions were swabbed and plated on chocolate agar (at 37 °C, 5% CO2, 18–72 h for growth rate variation for various isolates), blood agar (at 37 °C, 16–24 h) and MacConkey agar (at 37 °C, 16–24 h). The bacterial isolates were identified by colony morphology, Gram stain (Gram negative bacillus), nicotinamide adenine dinucleotide dependence (only growing on chocolate agar) and virulence-associated trimeric autotransporter group 3 colony PCR (Pina et al., 2009).

Molecular serotyping mPCR

The molecular serotyping assay for H. parasuis isolates was modified from a previously published method (Howell et al., 2015). The species-specific (sp-sp) amplicon was used as an internal control. A loopful of bacteria from a passaged plate of pure culture was resuspended in 30 μL ultrapure H2O, which was heated to 100 °C for 30 min then centrifuged at 4,000×g for 1 min. The supernatant was used in the mPCR reaction. Isolates from various lesions or pigs from the same herd were serotyped. If they belonged to same serovar, they were considered one isolate.

Each PCR reaction was performed in a total volume of 30 μL containing ultrapure H2O, 1 × DreamTaq buffer, 250 μM dNTP, 0.2 μM concentrations of forward and reverse serovar-specific primers, 0.04 μM concentrations of forward and reverse sp-sp primers, 1.25 U of DreamTaq DNA polymerase (Thermo Fisher Scientific, Waltham, MA, USA) and one μL of supernatant. The thermocycling conditions consisted of 5 min at 94 °C, 30 cycles of 30 s at 94 °C, 30 s at 58 °C and 1 min at 72 °C, and then a final extension at 72 °C for 5 min. The molecular serotyping mPCR amplicons were stained with ethidium bromide and analyzed using a 20 cm-long 2% agarose gel. A 50 bp DNA ladder RTU (GeneDireX, Las Vegas, NV, USA) and Bio-1D software (Vilber Lourmat, Collégien, France) were used to estimate molecular size. The electrophoresis conditions were an electric field six V/cm (300 V, 50 cm full-length electric field) and 3 h. The results were confirmed by twice repeating tests.

Sequencing and analysis of unexpected PCR-amplified products

Unexpected amplicons of the molecular serotyping mPCR products were cloned using a TA cloning kit (Yeastern Biotech Co., Ltd. Taipei, Taiwan) and sequenced using an ABI 3730XL DNA analyzer (Applied Biosystems, Foster City, CA, USA). Sequence data were analyzed using MEGA7 (Molecular Evolutionary Genetics Analysis Version 7.0) software and Basic local alignment search tool (BLAST) database.

Pathological examination

Cases of sick animals or fresh, complete carcasses were subjected to necropsy for gross morphological examinations and H&E staining. Histopathological examination focused primarily on meningeal, pleural, pericardial, peritoneal and synovial cavities of joints, and lungs. Typical meningeal gross lesions were characterized by yellow to white exudate accumulation in the subarachnoid space, on pia mater and in the sulci (Fig. S1). Meningeal histopathological lesions were principal neutrophils and few mononuclear inflammatory cells infiltration on pia mater with fibrin and cellular debris deposit (Fig. S2). Serosal lesions were characterized by yellow to white exudate accumulation in pleural, pericardial, abdominal and joint synovial cavities and yellow to white fibrin covering the visceral and parietal serosa (Figs. S3–S5). The histopathological lesions of serositis were principal neutrophils and few mononuclear inflammatory cells infiltration with fibrin deposit (Fig. S6). In typical cases, H. parasuis resulted in bronchopneumonia with numerous neutrophils, mononuclear inflammatory cells, erythrocytes, cellular debris and fibrinous exudate accumulation in alveoli (Figs. S7 and S8). Due to disease duration, lesions varied in field. Other lesions infected H. parasuis were also involved to determine pathological patterns, including chronic fibrous serositis with angiogenesis and mononuclear inflammatory cells infiltration (Fig. S9), and meningitis with principal macrophages infiltration (Fig. S10).

Detection of porcine reproductive and respiratory syndrome virus

Nucleic acid extraction of pulmonary tissue was performed on a MagNA Pure LC 2.0 by using the MagNA Pure LC total nucleic acid isolation kit (Roche Applied Science, Indianapolis, IN, USA). Following cDNA synthesis was using PrimeScript™ RT reagent kits (Takara, Kyoto, Japan). Porcine reproductive and respiratory syndrome virus (PRRSV) reverse transcription real-time PCR was performed as previously described (Lin et al., 2013).

Statistical analysis

Fisher’s exact test was used to compare the frequency of H. parasuis infected lesions and the percentage of various lesion patterns using GraphPad Prism software (GraphPad Software, La Jolla, CA, USA). Variables were considered significant at a 0.05 level (two-sided).

Results

H. parasuis isolates, origins and pathological lesion patterns

A total of 133 isolates of H. parasuis were isolated from August 2013 to February 2017. The isolates were taken from 277 lesions on 155 diseased animals from 124 infected herds. Isolates from a herd serotyped as a single serovar were calculated as one isolate. Of 155 H. parasuis cases, 12 cases (7.7%) belonged to suckling pigs (≤3-week-old), 133 cases (85.2%) belonged to nursery pigs (4- to 12-week-old), seven cases (4.5%) belonged to growing pigs (13- to 26-week-old) and one case belonged to a breeding boar. Age information for two cases was unknown. A total of 86 cases (55.5%) had H. parasuis isolated from lung lesions with or without serosal lesions.

A total of 108 animals were necropsied with complete pathological examination and further correlated to pathological pattern and isolation proportion (Table S3). Of the H. parasuis infected animals, 54.6 % had serositis and pulmonary tissue lesions, 41.7% had serosal lesions only and 3.7 % displayed only pulmonary lesions (Fig. 1).

Figure 1 Distribution of Haemophilus parasuis serovars according to lesion pattern.

Serositis only: animals were diagnosed with H. parasuis positive serosal lesions. Pulmonary lesion only: animals were diagnosed with H. parasuis positive pulmonary lesions. Data were analyzed by Fisher’s exact test and variables were considered significant at a 0.05 level (two-sided).

A total of 106 cases (98.1%) had bronchopneumonia, 64 cases (59.3%) displayed H. parasuis positive lung lesions. A total of 78 cases (72.2%) registered as positive for PRRSV via reverse transcription real-time PCR screening. The proportion of 204 H. parasuis infected lesions from 108 animals with complete pathological examination were meninges (10.3%), pleura (20.1%), pericardium (16.2%), peritoneum (13.7%), joint synovial cavity (9.3%) and lung (30.4%) (Fig. 2). The percentage of lung lesions showing H. parasuis infection was significantly higher than the percentage of serosal lesions (P < 0.05).

Figure 2 Haemophilus parasuis isolation proportion of 204 lesions of 108 pathological diagnosed cases.

Fisher’s exact test was used to compare the frequency of H. parasuis isolation lesions. P-value < 0.05 was considered a significant difference.

Serovar distribution by molecular serotyping assay

Of the 133 isolates, 91 (68.94%) isolates were typed using molecular serotyping mPCR. The most common serovars were serovar 5/12 (38.2%) and serovar 4 (27.5%) followed by serovar 14 (2.3%), serovar 1 (0.8%), serovar 2 (0.8%) and serovar 9 (0.8%) (Fig. 3). However, the product sizes of the serovar-specific primers analyzed by Bio-1D software were varied from the original publication (Howell et al., 2015). Furthermore, there were still 41 isolates (29.8%) classified as molecular serotyping non-typable (MSNT); these were divided into four groups based on the appearance of unexpected amplicons or the lack of serovar-specific amplicons. A total of 18 isolates (13%) only positive for a sp-sp marker were categorized as MSNT group 1. A total of 19 isolates (14.5%) were placed in MSNT group 2; these displayed amplicons of 300, 830 and 1,000 bp. Two isolates (1.5%) which showed unexpected amplicons at 500 and 660 bp were categorized as MSNT group 3. One isolate (0.8%), showing an amplicon of 300 bp was categorized as an MSNT group 4 isolate (Fig. 3; Figs. S11–S15).

Figure 3 Molecular serotyping results with or without sequence results for 133 Haemophilus parasuis isolates.

Identification of serovar-specific amplicons

The amplicons generated from molecular serotyping mPCR analyzed using Bio-1D software were not precisely the same sizes as previously indicated in the original description of this assay (Howell et al., 2015). The product size of a specific amplicon found in serovar 4 was mentioned at 320 bp in the original publication but the PCR generated an amplicon of nearly 350 bp which might be confused with serovar 6. In serovar 5, the PCR results generated an amplicon larger than 450 bp mentioned in the original publication which might be confused with serovar 7. The product size of serovar 9 serovar-specific primers, mentioned at 710 bp in the original publication, was smaller than the 700 bp ladder marker and might be confused with serovar 8. In light of these conflicting results, the isolates were serotyped again to confirm the sizes, and the amplicons were subsequently sequenced. Comparisons of the molecular serotyping original publication described, Bio-1D software analyzed, and BLAST product sizes are shown in Table 1 (Howell et al., 2015). The product sizes analyzed by Bio-1D software, BLAST and sequence are more consistent.

Table 1 Product size by molecular serotyping assay.

Gene	Serovar	Product size in the original publication (bp)	Product size (bp) predicted by BLAST	Aligned sequence accession number	Product size (bp) measured by Bio-1D software	Product size (bp) according to sequence	
funB	1	180	183	CL120103	184	183	
wzx	2†	295	294	CL120103	N/A	N/A	
glyC	3†	610	618	KC795327.1	N/A	N/A	
wciP	4	320	349	KC795356.1	350	349	
wcwK	5 or 12	450	468	KC795341.1	469	468	
gltI	6†	360	378	KC795372.1	N/A	N/A	
funQ	7†	490	499	CP009158.1	N/A	N/A	
scdA	8†	650	634	KC795411.1	N/A	N/A	
funV	9	710	676	KC795429.1	675	676	
funX	10†	790	784	KC795448.1	N/A	N/A	
amtA	11†	890	883	KC795474.1	N/A	N/A	
gltP	13†	840	836	KF841370.1	N/A	N/A	
funAB	14	730	710	KC795520.1	708	710	
funI	15†	550	550	KC795537.1	N/A	N/A	
HPS_219690793	All	275	276	CP020085.1	276	276	
Note:

† This serovar was not detected in this study.

The unexpected PCR products of the MSNT isolates were cloned for sequencing (Table 2). The MSNT group 2 amplicon was 297 bp; this product was generated with the serovar 13 specific forward and the serovar 14 specific reverse primers targeting gltP gene as a marker of serovar 13 in the polysaccharide biosynthesis locus. These primers were paired because the target sequences in the respective serovars shared homologous segments. The other PCR generated an amplicon product of the MSNT group 2 isolate determined to be 836 bp, and was identified as a serovar 13 specific product. The Bio-1D software measured a 500 bp product of the MSNT group 3 isolate as 499 bp; this amplicon was identified as a serovar 7 specific product. The 300 bp product found in the MSNT group 4 isolate, (sequencing results indicated it was 297 bp) was generated by pairing the serovar 13 specific forward primer with the serovar 14 specific reverse primers. This result was the same as that generated using the same primer pair of DNA isolated from MSNT group 2.

Table 2 Unexpected products of serotyping multiplex PCR.

Molecular serotyping non-typable isolate	Serovar according to sequence	Product size (bp) measured by Bio-1D software	Product size (bp) according to sequence	Amplified primer	
Group 1	Unknown†	None‡	None	None	
Group 2	Serovar 13	300	297	S13F, S14R	
830	836	S13	
1000	N/A§	N/A	
Group 3	Serovar 7	500	499	S7	
660	N/A	N/A	
Group 4	Serovar 13	300	297	S13F, S14R	
Notes:

† Serovar could not be defined without any serovar-specific product sequence result.

‡ There was no serovar-specific product.

§ Cloning of serovar-specific product was failed.

Serovar distribution based on molecular serotyping assay and sequencing

The molecular serotyping assay combined with sequencing results reduced the percentage of isolates classified as MSNT from 30.1% to 13.5% (Fig. 3). The dominant serovars were serovar 5/12 (37.6%), serovar 4 (27.8%) and serovar 13 (15%) followed by serovar 14 (2.3%), serovar 7 (1.5%), serovar 1 (0.8%), serovar 2 (0.8%) and serovar 9 (0.8%). Combining the sequencing results showed serovar 13 is a common serovar.

Relationship between pathological lesion patterns and serovars

The distribution of H. parasuis serovars in lesions from necropsied animals were serovar 5 (42.6%), serovar 4 (21.3%), serovar 13 (20.4%) and MSNT group 1 (13%). These categories were further subdivided into animals displaying both serosal and pulmonary lesions, those with only pulmonary lesions, and those with lesions found only in serosa. The respective percentages of lesions vs. serovars, and the pattern of lesions in infected animals were showed in Fig. 1 and Table S4. Necropsied animals with both serosal and pulmonary lesions were the most frequent; animals with pulmonary lesions alone were the least frequent (p < 0.0001). Serovars 4 and 5/12 showed similar results, the MSNT group 1 both serosal and pulmonary lesions were more frequent than serosal lesions alone. Serovar 13 had more serosal lesions than the combination of serosal and pulmonary lesions.

A nine herds (7.3%) had populations infected with two H. parasuis serovars. One herd contained a population with lesions infected with serovars 1 and 4. Three herds were infected with serovars 4 and 5. Serovar 5, 13 and 5, 14 co-infections were seen in single herds. The infected lesions were located in animals displaying a variety of tissue lesion patterns. Other four herds contained individuals co-infected with two H. parasuis serovars. One clinical case showed pleural and pulmonary lesions coinfected with H. parasuis serovars 4 and 7, respectively. A separate herd contained one case of pulmonary lesions with serovar 5, as well as pleural, pericardial and peritoneal lesions infected with H. parasuis serovar 13. One case was co-infected with serovars 5 (pulmonary) and 13 (pleura and pericardium). A fourth case showed coinfection with serovar 4 and an MSNT group 1 isolate taken from separate pulmonary lesions.

Discussion

This is the first study describing serovars of H. parasuis defined by molecular serotyping in Taiwan. The most common serovars are serovar 5/12, 4 and 13, followed by MSNT isolates. Even though serotyping assays vary, the serovar population profile of H. parasuis in Taiwan is similar to profiles described in several other studies (Table S1) (Angen, Svensmark & Mittal, 2004; Blackall, Rapp-Gabrielson & Hampson, 1996; Cai et al., 2005; Castilla et al., 2012; Del Rio, Gutierrez & Rodriguez Ferri, 2003; Dijkman et al., 2012; Howell et al., 2015; Kielstein & Rapp-Gabrielson, 1992; Luppi et al., 2013; Ma et al., 2016; Oliveira, Blackall & Pijoan, 2003; Rapp-Gabrielson & Gabrielson, 1992; Rubies et al., 1999; Tadjine et al., 2004). Most commercial H. parasuis vaccines are inactivated vaccines, which provide protection against the same serovar but are unable to provide protection from challenge using different serovars (Miniats, Smart & Rosendal, 1991b; Nielsen, 1993; Smart & Miniats, 1989; Takahashi et al., 2001). Candidate serovar composition in H. parasuis vaccine determines the success of a vaccine strategy against H. parasuis (Takahashi et al., 2001). Therefore, the distribution of serovars in herds is an important factor in outlining vaccination strategies and vaccine developments aimed at the prevention and control of Glässer’s disease.

Indirect hemagglutination assay was applied to H. parasuis serovar differentiation to decrease the proportion of H. parasuis isolates classified as CSCR (Cai et al., 2005; Del Rio, Gutierrez & Rodriguez Ferri, 2003). De-encapsulation due to multiple passages results in non-reaction with antisera and cross reactivity of isolate antigens to diagnostic (immune-based) test reagents are the primary factors behind CSNT and CSCR H. parasuis isolates, respectively (Cai et al., 2005; Kielstein & Rapp-Gabrielson, 1992; Oliveira, Blackall & Pijoan, 2003; Rapp-Gabrielson & Gabrielson, 1992; Turni & Blackall, 2005). The presence of CSNT and CSCR isolates confounds epidemiological surveys used to assess H. parasuis isolate population profiles, and impairs efforts to generate effective vaccines against this pathogen. The correlation between the capsule and serovar of H. parasuis is well established; a multiplex serotyping PCR was developed with this in mind (Howell et al., 2015, 2013). This protocol can be employed to type isolates previously classified as CSNT and CSCR via traditional (immunological) methods. The mPCR serotyping reduced the incidence of CSNT and CSCR H. parasuis but does not completely eliminate the issue of CSNT and CSCR isolates. One reason may be the sequence similarity of different serovar-specific primers and serovar-specific products. Another factor may be deletions and/or unknown sequences within certain antigenic markers (Ma et al., 2016). This underscores the importance of MSNT isolate whole-genome sequencing for in silico serotyping and improving the molecular serotyping assay. Emergence of MSNT isolates by the molecular serotyping assay may be due to insufficient or incomplete sequence data for H. parasuis from Asia. When this assay was developed, there were only nine Asian H. parasuis isolates in a 212-isolate database (seven from Japan, two from China). Investigating the sequences and gene composition of Asian H. parasuis isolate capsule loci may be key for assaying and serotyping MSNT isolates. Besides, absence of serovar-specific markers in polysaccharide biosynthesis loci in MSNT isolates may create antigenic variation impairing vaccine strategies. Therefore, it is also important to study the antigenic variation due to gene mutation and/or absence in polysaccharide biosynthesis loci in the future.

Thus far, molecular serotyping has been challenging as there are 15 serovars, making it difficult to design serovar-specific primers yielding differential results. Some primer pairs produce amplicons from different H. parasuis serovars that vary by less than 20 bp-a difference that is hardly detected especially when the amplicon size is larger than 600 bp. In our study, electrophoresis using longer agarose gels was performed to enhance the ability of the procedure to discriminate DNA fragment sizes. Bio-1D software was applied to more accurately measure product size based on the intensity of the bands and decrease human operation error. In the case of molecular serotyping tests resulting in ambiguities, serovar-specific primer pairs may be used (in simplex PCR format) to confirm or classify hard-to-identify serovars. According to sequence analysis, the product sizes described in the original publication were not accurate (Howell et al., 2015). The corrected product sizes are important to avoid mis-serotyping.

According to a previous study, pigs were infected with H. parasuis serovars 1, 5, 10, 12, 13 and 14 showed high mortality. Pigs challenged with serovars 2, 4, 8 and 15 showed polyserositis. Pigs inoculated with serovars 3, 6, 7, 9 and 11 resulted in no clinical symptoms or lesions (Kielstein & Rapp-Gabrielson, 1992). Serovars 5/12, 4, 13 and 7 are the most common serovars in most countries worldwide (Table S1). A previous study showed that serovar 4 and 13 have a higher prevalence in systemic infection than in only respiratory disease (Luppi et al., 2013). Our data also showed similar results of serovar 4, 5/12, 13 and MSNT isolates. There may be some correlation between serovar and virulence because serovars are defined by capsule which can directly interact with host cells and has been proven to be a key virulence factor relating to phagocytosis resistance (Olvera et al., 2009). Besides, it should be considered if the impact of some isolates resulting in only pulmonary lesions are underestimated in field due to absence of serositis and typical pulmonary lesions. The role and economic impact of H. parasuis in pulmonary infection animals related to porcine respiratory disease complex in field is also worthy of investigation in the future. In our study, serovars 7 and 9 caused serositis with or without respiratory lesions. A serovar 5 isolate was isolated from an animal with only bronchopneumonia lesions and another with lesions in both the serosa and lung tissues in the same herd. Therefore, the results show clinical manifestations of Glässer’s disease are influenced by multiple factors, including host, stress, environment, co-infection with different serovars or other pathogens and gene differences between infecting H. parasuis isolates (Boerlin et al., 2013; Howell et al., 2014; Li et al., 2009). In general, most Glässer’s disease cases in nursery pigs were co-infected with PRRSV in our data. This may be because PRRSV can cause immunosuppression by reducing non-specific bactericidal activity of pig alveolar macrophages and stimulating interleukin-10 production, which down-regulates inflammatory cytokines (Drew, 2000; Flores-Mendoza et al., 2008; Suradhat & Thanawongnuwech, 2003). The previous studies showed PRRSV does not result in increased Glässer’s disease by experimental challenge (Segales et al., 1999; Solano et al., 1997). However, significant association between H. parasuis and PRRSV in field was reported (Palzer et al., 2015). Recently studies also showed PRRSV can induce bronchopneumonia with Bordetella bronchiseptica which is a part of normal upper respiratory microbiota and predispose to colonization with H. parasuis (Brockmeier, 2004; Brockmeier et al., 2001). Co-infection of pig alveolar macrophages with PRRSV and H. parasuis leads to pro-inflammatory mediated immunopathology by synergistic effect (Kavanova et al., 2015; Li et al., 2017). In the future, the synergistic effect between PRRSV and H. parasuis resulting in economic losses in field is worthy of further investigation. Other factors also interact with H. parasuis including the stress of weaning and maternal antibody reduction. However, highly virulent H. parasuis isolates might be considered primary pathogens (Aragon, Segalés & Oliveira, 2012). In our study, some H. parasuis isolates caused serositis and sudden death without co-infection, even in growing pigs and breeding boars.

A previous study showed H. parasuis can access the blood stream through invasion of the mucosal surface in the nasal cavity (Vahle, Haynes & Andrews, 1997). In our study, pulmonary lesions showed higher pathogenic H. parasuis infection rates than serosal lesions. These results are in accordance with a previous study from the Netherlands (Dijkman et al., 2012). H. parasuis invasion and survival in lung tissue is likely a key feature for the onset of disease (Olvera et al., 2009; Vahle, Haynes & Andrews, 1995). Our results show H. parasuis infected animals with lesions found in dual anatomical locations (pulmonary and serosal) occur at a higher rate than infected animals with lesions located in only one tissue type. A previous study also mentioned lung is one of the most successful sites for acute (serovar 12) and subacute (serovar 4) isolate recovery (Turni & Blackall, 2007). Therefore, lung is an important origin for H. parasuis isolation and a target organ for Glässer’s disease diagnosis. Pulmonary infections may be an important step for H. parasuis systemic infections.

Others have reported isolation of multiple isolates from single pig farms (Cerda-Cuellar et al., 2010; Oliveira, Blackall & Pijoan, 2003; Olvera, Calsamiglia & Aragon, 2006a; Olvera, Cerda-Cuellar & Aragon, 2006b). Our results also show different serovars cause disease in a single herd, or even in a single animal, although the latter scenario is fairly uncommon. In most situations, Glässer’s disease is caused by one isolate (Rafiee et al., 2000), but several isolates may be present at a given farm (Turni & Blackall, 2010). Therefore, it would be useful to develop universal or multivalent capsular vaccines against multiple serovars. The possibility of cross talk between different pathogenic H. parasuis isolates at a given site may be worthy of investigation.

Conclusions

The dominant serovars of H. parasuis in Taiwan are serovars 5/12, 4 and 13, followed by MSNT isolates. Proportions of isolates in those serovars resulting in both serosal and pulmonary lesions are significantly higher than pulmonary lesion. Pulmonary lesions may be most important for H. parasuis isolation and may serve as points of origin for systemic H. parasuis infections in hosts.

Supplemental Information

Supplemental Information 1 Gross meningeal lesion in H. parasuis infected pigs.

Click here for additional data file.

Supplemental Information 2 Histopathological suppurative meningitis lesion in H. parasuis infected pigs.

Click here for additional data file.

Supplemental Information 3 Gross pleural and peritoneal lesions in H. parasuis infected pigs.

Click here for additional data file.

Supplemental Information 4 Gross epicardial lesion in H. parasuis infected pigs.

Click here for additional data file.

Supplemental Information 5 Gross joint synovial cavity lesion in H. parasuis infected pigs.

Click here for additional data file.

Supplemental Information 6 Histopathological fibrinous serositis lesion in H. parasuis infected pigs.

Click here for additional data file.

Supplemental Information 7 Gross lung lesion in H. parasuis infected pigs.

Click here for additional data file.

Supplemental Information 8 Histopathological pulmonary lesion in H. parasuis infected pigs.

Click here for additional data file.

Supplemental Information 9 Histopathological fibrous serositis lesion in H. parasuis infected pigs.

Click here for additional data file.

Supplemental Information 10 Histopathological meningitis lesion in H. parasuis infected pigs.

Click here for additional data file.

Supplemental Information 11 Band patterns of molecular serotyping mPCR for Haemophilus parasuis.

Lane M: 50 bp DNA Ladder, lane S5: serovar 5 or 12, lane G2: molecular serotyping non-typable group 2, lane S4: serovar 4, lane S9: serovar 9, lane G1: molecular serotyping non-typable group 1, lane NC: negative control.

Click here for additional data file.

Supplemental Information 12 Band patterns of molecular serotyping mPCR for Haemophilus parasuis.

Lane M: 50 bp DNA Ladder, lane S4: serovar 4, lane S5: serovar 5 or 12, lane G2: molecular serotyping non-typable group 2, lane S14: serovar 14, lane G1: molecular serotyping non-typable group 1, lane NC: negative control.

Click here for additional data file.

Supplemental Information 13 Band patterns of molecular serotyping mPCR for Haemophilus parasuis.

Lane M: 50 bp DNA Ladder, lane S5: serovar 5 or 12, lane S4: serovar 4, lane G1: molecular serotyping non-typable group 1, lane G2: molecular serotyping non-typable group 2, lane NC: negative control. Histopathological bronchopneumonia lesion in H. parasuis infected pigs.

Click here for additional data file.

Supplemental Information 14 Band patterns of molecular serotyping mPCR for Haemophilus parasuis.

Lane M: 50 bp DNA Ladder, lane G1: molecular serotyping non-typable group 1, lane S4: 608 serovar 4, lane S5: serovar 5 or 12, lane S14: serovar 14, lane G4: molecular serotyping non- 609 typable group 4, lane G3: molecular serotyping non-typable group 3, lane NC: negative control.

Click here for additional data file.

Supplemental Information 15 Band patterns of molecular serotyping mPCR for Haemophilus parasuis.

Lane M: 50 bp DNA Ladder RTU (GeneDireX), lane G2: molecular serotyping non-typable group 2, lane S4: serovar 4, lane S5: serovar 5 or 12, lane G1: molecular serotyping non-typable group 1, lane G3: molecular serotyping the non-typable group 3, lane NC: negative control.

Click here for additional data file.

Supplemental Information 16 Serovar distribution of Haemophilus parasuis in different countries or regions by different serotyping assays.

Click here for additional data file.

Supplemental Information 17 Raw data.

Click here for additional data file.

Supplemental Information 18 Cases distinguished by complete pathological examination or not.

Click here for additional data file.

Supplemental Information 19 Relationship between pathological diagnoses and serovers.

Click here for additional data file.

We appreciate the following people for their help with this study: swine veterinarians at the Animal Disease Diagnostic Center for assistance in necropsy, Ying-Xiu Lian for assistance in viral nucleic acid detection and Qiong-Yi Huang to demonstrate bacterial preservation. Swine veterinarians who assisted in this study: Guan-Shiuan Su, Chih-Chung Chang, Shu-Wei Chang, Kuan-Lin Li, Ling-Fang Wang, Guang-Ting Tsai, Ni-Jyun Ke, Ting-Han Lin, Sheng-Yuan Wang, Hong Liu, Jia-Liang Hong, Joan Wang and Yu-Hsuan Chen.

Additional Information and Declarations

Competing Interests

Author Contributions

Animal Ethics

Data Availability

The authors declare that they have no competing interests.

Wei-Hao Lin conceived and designed the experiments, performed the experiments, analyzed the data, prepared figures and/or tables, authored or reviewed drafts of the paper.

Hsing-Chun Shih performed the experiments.

Chuen-Fu Lin contributed reagents/materials/analysis tools.

Cheng-Yao Yang contributed reagents/materials/analysis tools.

Yung-Fu Chang contributed reagents/materials/analysis tools.

Chao-Nan Lin conceived and designed the experiments, contributed reagents/materials/analysis tools, prepared figures and/or tables, authored or reviewed drafts of the paper, approved the final draft.

Ming-Tang Chiou conceived and designed the experiments, contributed reagents/materials/analysis tools, prepared figures and/or tables, authored or reviewed drafts of the paper, approved the final draft.

The following information was supplied relating to ethical approvals (i.e., approving body and any reference numbers):

The study did not involve any animal experiment. The Institutional Animal Care and Use Committee (IACUC) of National Pingtung University of Science and Technology did not deem it necessary for this research group to obtain formal approval to conduct this study.

The following information was supplied regarding data availability:

The raw data are provided in Table S2.

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
