# Peer review of "Molecular serotyping of Haemophilus parasuis isolated from diseased pigs and the relationship between serovars and pathological patterns in Taiwan"

_PeerJ, doi:10.7717/peerj.6017_

## Round 0.1 · original submission · Major Revisions

· Academic Editor

Major Revisions

We have received three reviews for your paper. Two of the three suggest revisions and in considering the comments of all reviewers, I think a revision is in order, but the paper is publishable. All reviewers felt the data were good and the paper useful to the community. Two of the three provide clear details and ways to improve the methods and discussion and to strengthen the text. I encourage you to work through these reviews carefully and submit your revision.

·

Basic reporting

This manuscript describes pathology and serotyping of Glassers Disease in Taiwan. The text can be confusing and needs clarification especially when talking about diagnosis, identification and serotyping. (L 32)
Histopathology is mentioned but lesions are not described.
Diagnosis of other diseases section should be reduced to a few lines as it isn't a focus of the paper. The discussion states many animals were co-infected with PRRSV but there is little discussion in results. what is the significance of this co-infection?
L 169 Here and in other areas of the text where you present case, isolate, lesion, or serotype data the numbers often don't agree with subsequent paragraphs or tables/figures. Please clarify these sections and insure the data agree with tables and figures.
L85-190 The information in lines 185-190 is very confusing
L219-220 Was this in a multiplex reaction? Was the amplicon generated in the polysaccharide operon.
Early in the manuscript you say 5/12 was not typeable using molecular testing but later show it as the most common serotype found. Need to clarify how it was typed.
L241 refers to a table and a figure but the data don't seem to agree. suggest you use one or the other but not both and make sure the numbers match
L 248-251 number of infected herds vary. They should be the same.
L310 was the study a pig infection model? Clearly identify your work from others-it gets a bit confusing which work is being referred to in this section.

Experimental design

Aims and scope are identified
Research question is relevant to understanding the epidemiology of Glasser's disease and the methods used for serotyping

The aims state a desire to correlate pathology with serotype but pathology is limited to gross examination and H&E stains. Special stains for demonstrating bacteria could complement this effort. pathology examination was only used to identify lesions in tissue

I didn't see any use of known serotypes for generation of amplicons or gel-diffusion assays. Positive controls would be a good addition to the data.

Validity of the findings

Results of molecular serotyping were found to differ from the original descriptions of the assays. Differences in amplicon size were noted and resolved by sequencing.
Nontypeable organisms were identified using PCR and were subsequently typed using sequencing in some instances but in others they remained un-typeable. It was unclear if any other serotyping methods were used with these organisms. Also, the significance of finding nontypeable isolates by PCR but typeable by sequencing and how this might impact vaccine strategies was not discussed.

Additional comments

This manuscript has a lot of data in it and the way it is presented can be confusing in some of the sections. Try not to put to much data in one sentence or paragraph. Use tables to show data relationships and perhaps reduce the text by referring to the table/figure.

·

Basic reporting

Clear and unambiguous, professional English used throughout, with some typos.
Literature references, sufficient field background/context provided.
Self-contained with relevant results to hypotheses.

Experimental design

Clear and answer all the hypothesis of the study.

Validity of the findings

Conclusion are well stated, linked to original research question & limited to supporting results.

Additional comments

This study was well designed and answered all the questions the authors interested. Good plan of study.

·

Basic reporting

This manuscript describes molecular serotyping of “Haemophilus parasuis isolated from diseased pigs and the relationship between serovars and pathological patterns in Taiwan”. This research work also focus on rectify the PCR amplicons by Howell (2015), “Development of a multiplex PCR assay for rapid molecular serotyping of Haemophilus parasuis”. This work will help other researchers to identify the PCR amplicon more accurately. In general, the studies are informative and provide new knowledge to the readers. However, the discussion portion is a bit weak and few essential issues need to be further addressed.

Experimental design

The experiment is well design and good.

Validity of the findings

The author need to emphasis the important to differentiate the PCR amplicon, and how the new findings will be useful for researchers for the multiplex PCR work.
The relationship between serovars and the pathological patterns were not strong.

Additional comments

This manuscript describes molecular serotyping of “Haemophilus parasuis isolated from diseased pigs and the relationship between serovars and pathological patterns in Taiwan”. This research work also focus on rectify the PCR amplicons by Howell (2015), “Development of a multiplex PCR assay for rapid molecular serotyping of Haemophilus parasuis”. This work will help other researchers to identify the PCR amplicon more accurately. In general, the studies are informative and provide new knowledge to the readers. However, the discussion portion is a bit weak and few essential issues need to be further addressed.
Introduction
Line 90 – Please input a statement about prevalence rate of H. parasusi and its estimation economic value in Taiwan

Materials and Methods
Line 100 – Please explain what are the synovial cavities refer to? Joints?

Result
Line 182 – Please use percentage for Pasteurella multocida also (all the diseases were in percentage expect this)
Line 183 – the Salmonellosis was referring to systemic disease or diarrhea episode? Please specific
Line 207-209 – Please input few statements to justify why it is important to differentiate the PCR amplicon, and how the new findings will be useful for researchers? For example, PCR amplicons for Serovar 4 (350 bp) and Serovar 6 (360 bp) were too close and easily confuse. PCR amplicons for Serovar 5 (<500 bp) and Serovar 7 (500 bp) can’t be differentiate.
Line 210 – Base on the supplementary figure 5, the reviewer disagree with this statement “The product size of serovar 9 serovar-specific primers, predicted at 710 bp, was smaller than the
700 bp ladder marker”. Please remove or rephrase it.

Discussion
Line 285 - “CSNT and CSCR isolates have recently been serotyped using molecular serotyping in the UK (Howell et al. 2015)”. This sentence is redundant, please remove it.

Line 286 - “However, even though 7 mPCR serotyping has reduced the incidence (percentage) of CSNT and CSCR H. parasuis isolate, some isolates still yielded ambiguous PCR results”. Please rephrase the sentence as a sentence can’t start with both “however” and “even though”.

Line 288 –Please remove “As a result”
Line 294 – “Gaps in the molecular serotyping assay may be the result of insufficient or incomplete sequence data for H. parasuis from Asia”. This sentence is incomplete, please rephrase it.

Line 300 – Please remove “detectable using a gel”
Line 302 – “most agarose gels cannot detect with accuracy”, please rephrase this sentence

The result for relationship between serovars and the pathological patterns were not strong.

---

## Round 0.2 · accepted · Accept

· Academic Editor

Accept

Your paper is much improved and is nearly ready for publication. While I am accepting it, please note, I would like you to address the comments from the re-review provided. There are several typos to adjust and some minor considerations to strengthen the paper. I am accepting this paper on good faith that you will address these points in a final submission. Nice work.

# ·

Basic reporting

The manuscript is much improved and language is generally quite clear. However there are two sentences needing rewording for clarity. In both the pre- abstract and abstract lines 30-31 the manuscript says types 5/12 are not typeable when what you mean is they are not distinguishable by molecular typing. Please consider making this change

I indicated some omission suggestions to improve the manuscript. See first line of results in the abstract (remove "that") and line 5 of abstract results (remove "Futhermore")
Line 2 of Discussion there is a space between i and solates -close to make "isolates"
Line 308 "underestimated"
Line 322 remove "have" and "an"

Experimental design

Experimental design is much better defined, methods are detailed and supported

Validity of the findings

Multiple strains of H. parasuis are identified and some previously shown to be nonpathogentic are now associated with disease. This is an important finding and suggests serotype or molecular type may not correlate well with pathogenicity. Good job on identifying new disease associations.

Additional comments

Finally, related to vaccine production, you identified types 5/12, 13, 4 and 7 as the dominant types causing disease in pigs in Thailand. Since the vaccine is a killed vaccine and capsule is easily stripped from bacteria could a multivalent capsular vaccine be made with capsule from these molecular types? Or maybe a multivalent vaccine of 2-3 Molecular types could be generated specifically for piglets, or more mature pigs or herds. Just a thought you may want to consider in your discussion.